# Participatory practices at work change attitudes and behavior toward societal authority and justice

Sherry Jueyu Wu [1✉] & Elizabeth Levy Paluck[2]

Generalized attitudes toward authority and justice are often conceptualized as individual differences that are resistant to enduring change. However, across two field experiments with Chinese factory workers and American university staff, small adjustments to people's experience of participation in the workplace shifted these attitudes one month later. Both experiments randomly assigned work groups to a 20-minute participatory meeting once per week for six weeks, in which the supervisor stepped aside and workers discussed problems, ideas, and goals regarding their work (vs. a status quo meeting). Across 97 work groups and 1,924 workers, participatory meetings led workers to be less authoritarian and more critical about societal authority and justice, and to be more willing to participate in political, social, and familial decision-making. These findings provide rare experimental evidence of the theoretical predictions regarding participatory democracy: that local participatory experiences can influence broader democratic attitudes and empowerment.

[1] Collins Center A-415, Anderson School of Management, University of California Los Angeles, Los Angeles, CA 90095, USA. [2] 420 Peretsman Scully Hall, Department of Psychology, Princeton University, Princeton, NJ 08544, USA. ✉email: sherry.wu@anderson.ucla.edu

I n an attempt to understand the psychologies of fascist regime followers and of racism in the United States, psychologists following World War II started a distinguished line of work on generalized attitudes toward authority and justice[1]. Authoritarianism, a tendency to be deferent to authority and to be intolerant of deviance from existing social hierarchies[1–3], was conceptualized as a durable personality trait and as a syndrome that unifies interrelated attitudes toward authority, justice, and hierarchical relations[1,4]. Authoritarianism is believed to be in part heritable from parents, but also shaped by accumulated social experience and political context over time[3,5].

The present paper investigates, by contrast, whether authoritarianism, or generalized attitudes toward authority and justice, can change over the short term. To test this, we modified the dynamics of weekly staff meetings at two very different workplaces, transforming the meetings into participatory events where staff are encouraged to talk and supervisors mandated to listen. These participatory meetings took place once per week for 20 minutes at a time, over the course of six weeks. We randomly assigned this participatory meeting schedule to some work groups and not others, and compared workers' attitudes toward generalized authority and justice two to four weeks after the intervention ended. Unlike previous research, this approach allows us to determine whether conditions relevant to authority and hierarchy within a local group could causally influence attitudes toward a much broader societal context.

Our research question is motivated by perspectives on the development of generalized attitudes toward authority and justice that have received less attention in psychology. First, the research is motivated by Pateman's[6] theory of participatory democracy, which posits that workplaces inviting more worker participation can empower workers—decreasing workers' blind trust in authority and justice and motivating civic and political participation. Second, the research is motivated by Lewin's[7] idea that meaningful social groups can serve as "cultural islands," where local attitudes can develop from immersive group experiences, even when these experiences stand in contradiction to the group's broader societal context[7].

Below, we briefly review classic theories about the development and stability of authoritarian attitudes, including trust in authority and belief in justice. We then develop our hypothesis that short-term and immersive experiences, particularly within small groups, can influence generalized attitudes toward authority and justice in the larger society. Throughout, we treat attitudes toward authority and toward justice as separate but related attitudes[8,9].

*Authoritarianism: A stable trait, shaped by long-term experience.* Rooted in psychoanalytic theories, research on authoritarianism as a personality disposition[1,10] claims that authoritarianism emerges early in life and is linked to an avoidant attachment style[11]. This research suggests that it is a durable trait, and inheritable across generations[5], citing strong correlations between authoritarianism levels of young adults and their parents[12].

Subsequent research also suggested structural correlates to authoritarian attitudes[13,14]. For example, increased perception of societal-level threats, such as an economic downturn and elevated fear of crime, is positively associated with authoritarian attitudes[3,13,15]. Stable authoritarianism is distinct from attitudes toward a concrete authority figure or institution. There is ample evidence showing that attitudes toward a particular authority figure or institution can be shifted with situational interventions[16–19], whereas research shows the stability of generalized authoritarian attitudes over the lifespan. In this way, theories from psychology concur with theories from political science[20,21], that generalized attitudes toward authority and justice are shaped

by a prolonged experience of learning and socialization. As part of a generalized ideology or "syndrome," these attitudes develop from the breadth of a person's experience, including age, education, and social class, and are motivational in nature[13,22,23].

Ideas about the roots of an individual's authoritarianism can be traced back further to the work of Adam Smith and Karl Marx. Both scholars accentuated the role of long-term social experience, in particular the organization of daily work, which gradually shapes attitudes toward generalized authority and justice[24,25]. Using their work, later theorists argued that the lower socioeconomic groups like factory workers were "trained to subservience" during the course of their lifetime occupation, since it is among this group that authoritarian personalities are most frequently found[20].

None of these theories suggest that generalized attitudes toward authority and justice can be changed over the short-term. Rather, this body of work predicts that generalized authority and justice attitudes change in light of perceptions of a large societal threat, or from long-term experience with one's family, social and economic status, and occupation. However, a separate area of theoretical work, also focused on the role of experience, suggests that generalized authority and justice attitudes can be shaped by the structure of specific social contexts (i.e., the workplace) over a relatively shorter term.

*Local work groups: A training ground for social attitudes.* Influenced by Rousseau[26] and writings within political philosophy, the political scientist Carole Pateman[6] theorized that participatory experience within one's daily occupation educates and socializes individuals to have more "political efficacy," which in part translates to less deferent attitudes toward authority and toward existing systems of justice. Like the economic and political theories reviewed above, Pateman acknowledges that workplaces are important training ground for the development of these generalized attitudes, since they force individuals to spend most of their time in relationships of superiority and subordination. Based on her theory that local social structure has significant impact on individual "psychological qualities"[6], she predicts that workplaces that invite workers to participate in decision-making and management processes can affect workers' longstanding attitudes and even personality traits.

In psychology, classic theorizing by Kurt Lewin suggested that groups could create "cultural islands" by creating their own reality from their members' shared strong immersive experience within the group[7,27]. He explored this theory by studying a group of factory workers who were engaged over a number of weeks in more "democratic" working procedures—participatory and bottom-up, as opposed to autocratic and top-down. Lewin and colleagues[7] tracked workers' positive behavioral response to this intervention, but never theorized whether individuals could leave these cultural islands and shift their attitudes outside of the local workplace.

Theoretical predictions about the political influence of a participatory workplace have been empirically explored by political scientists and organizational scientists, using observational methods. Specifically, by surveying workers in companies about their perceived experience of participatory workplace practices, those studies provided mixed support for the hypothesis that perceived participation at work is correlated with more "democratic" workers who do not automatically defer to authority and current hierarchical arrangements[28–32]. Studies of the hypothesis of participatory democracy are also focused on workplaces in the United States and Europe[33]. Little research has been conducted in non-Western societies, especially those that are subject to non-democratic governments. To our knowledge, the central question of whether local participatory practices can cause changes in attitudes toward societal authority and justice has not

been tested with experimental methods, or in a broad range of settings.

In this research, we predict that individuals will become less deferent to generalized authority following an immersive group experience in which individuals are encouraged to speak up in front of authority and to assume more authority over their work life. The theoretical intuition is that the feeling of decreased deference will spill over into their assessment of other types of authority. Individuals may increase their appreciation for the specific authorities who facilitate this participatory experience, following the predictions of procedural justice[16], but more generally face authorities with greater skepticism or personal empowerment. Likewise, individuals may be less likely to believe in a just world and in hierarchical arrangements of social groups (e.g., workers and managers), if the fair and egalitarian procedures of their workplace experience provide a contrasting reference point to individuals' perception of fairness and equality in the world more generally.

We test these theoretical intuitions with field experiments manipulating work groups' increased participation and measuring subsequent attitudes toward societal authority and justice. Using a classic paradigm of participatory group meetings in social psychology[7], we experimentally manipulated 20 minutes of work groups' regular meeting time, once per week for six weeks. Weeks after the intervention ended, we measured workers' generalized attitudes toward authority and justice, and a cluster of related attitudes such as perceptions of hierarchy and of relationships between lower and higher status groups as well as self-reported participation behavior in politics.

Our first experiment was set in China, which is a particularly interesting site for testing the hypothesis that groups can become "cultural islands" when the broader environment does not support the group practices[7]. In China and in the factory where we worked, authorities are less likely to endorse democratic and decentralized practices[34,35]. Moreover, the participants in that experiment—young and less-educated female factory workers— are also on average less empowered to exercise or critique authority in their social and political contexts. Thus, Study 1 serves as a strong test of our hypotheses that participatory work contexts can change attitudes toward societal authority and justice, and that these shifts in attitudes can endure beyond the immediate group context. We conducted the second experiment in the United States with educated university administrative staff, in order to replicate and to test the generality of Study 1's conclusions.

Below, we report how we determined our sample size, all data exclusions (if any), all manipulations, and all measures in the study. We pre-registered all survey items, item groupings, and analyses at the Open Science Framework (https://osf.io/d9fnh/).

## Results

**Analysis strategy.** We tested the average treatment effects of participatory meetings on workers' attitudes a full month after the intervention had ended. Linear regressions used fixed effects for the seven departments in which the 65 groups were nested, a dummy variable indicating treatment, and a vector of pre-treatment individual demographic covariates to improve efficiency. Robust standard errors clustered by group accounted for residual covariance on the group level. Thus, to estimate the average treatment effect for an individual worker $i$ of group $j$,

$$Y_{ij} = \beta_0 + \beta_1 D_{ij} + \mathbf{g_1 Z_{ij}} + g_j + m_{ij}. \quad (1)$$

The regression coefficient $\beta_1$ represents the average treatment effect of the participatory meetings on worker attitudes, as measured by $Y_{ij}$ in self-report surveys four weeks after the end of

the experiment. $D_{ij}$ refers to a binary variable of experimental manipulation randomly assigned to the participants, in which $D_{ij} = 1$ refers to participatory meetings treatment condition and $D_{ij} = 0$ refers to the observer control condition. $\mathbf{Z_{ij}}$ is a vector of individual-level worker characteristics that are unaffected by the treatment (i.e., age, gender, marital status, and rural or urban origin). $g_i$ denotes a departmental fixed-effect, and $\mu$ is a zero-mean error term, assumed to be mutually independent across (but not within) groups.

As a robustness check, we also treat the work group as one unit ($N = 65$) by calculating group means of each outcome variable measured in the survey, and conduct between-group $t$-tests of a significant difference between participatory meetings condition and observer condition (results are consistent with what we report from linear regressions with fixed effects; see Supplementary Note 4). Because we estimate several outcomes from the survey data, we used a joint significance test against the null that none of the coefficients on treatment effects from multiple regressions are significant. As predicted, there was a jointly significant difference of the average treatment effects between workers in the participatory meetings condition and observer condition, $F_{1,58} = 8.06$, $p < 0.001$. Post-hoc power analysis indicates that the achieved power given our sample size and average effect size was 0.99. As another robustness check, we analyze individual survey items as outcomes (in addition to the pre-registered composite index scores; see Supplementary Tables 4–7). We report in the text only when results are not consistent for each of an index's individual items.

**Attitudinal changes toward authority and justice in study 1.** *Generalized attitudes toward authority*: The mean score of generalized attitudes toward authority for the whole sample of 1752 Chinese factory workers was 4.05 ($SD = 0.37$). This value indicates that, on average, workers tended to "slightly agree" with statements asserting complete obedience and respect for authority without question. However, as hypothesized, workers in the participatory meetings condition reported significantly lower scores in generalized attitudes toward authority ($M = 3.87$, $SD = 0.32$) than workers in the observer condition ($M = 4.23$, $SD = 0.33$, $\beta = -0.39$, $CI = [-0.55, -0.23]$, $SE = 0.08$, $p < 0.001$) (Fig. 1). Participatory meetings changed participants' generalized attitudes toward authority such that treatment workers registered as less authoritarian on a traditional scale of authoritarianism.

Because authoritarian attitudes are motivational in nature, meaning individuals who are high in authoritarianism may not wish to join participatory groups, one might expect a heterogeneous effect that participants who had higher baseline authoritarianism might be less responsive to the intervention. But we do not find a heterogeneous effect using proxy variables such as gender and age for authoritarianism.

*Belief in a just world*: For generalized attitudes and perceptions in justice, the mean score for the whole sample was 3.98 ($SD = 0.23$). This value indicates that, on average, workers tended to "slightly agree" with statements asserting belief in a just world. As hypothesized, workers in the participatory meetings condition reported significantly lower belief in a just world ($M = 3.86$, $SD = 0.22$) than workers in the observer condition who on average slightly agree with a just world belief ($M = 4.10$, $SD = 0.16$; $\beta = -0.26$, $CI = [-0.34, -0.18]$, $SE = 0.04$, $p < 0.001$, Fig. 1).

*Perceived intergroup conflict*: Participants in the treatment and control group did not differ in their perceptions of conflict between rich and ordinary people ($M_{PM} = 3.56$, $SD = 0.22$; $M_O = 3.50$, $SD = 0.23$; $p = 0.24$, n.s.), or between the capitalists and the

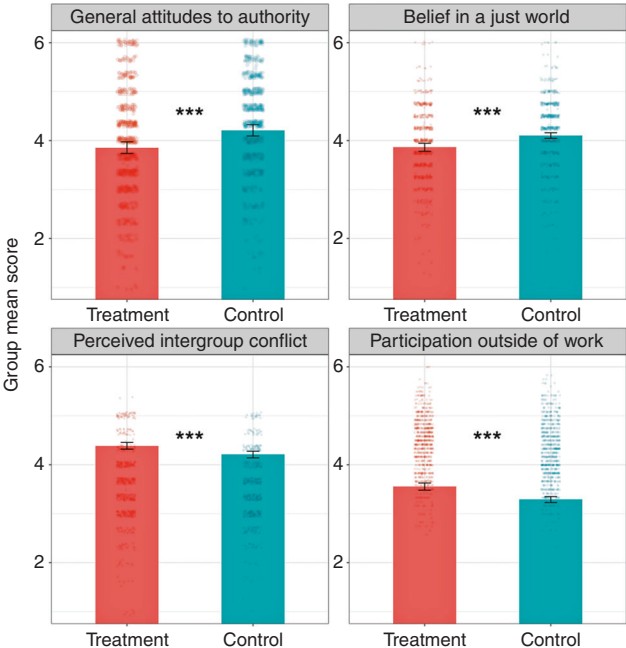

**Fig. 1 Study 1 group mean ratings.** Study 1 group mean ratings of general attitudes toward authority, belief in a just world, perceived intergroup conflict between managers and workers in the larger society, and participation outside of work, on a 6-point scale. Ratings are shown as a function of assignment to participatory meetings condition (treatment). Error bars show 95% confidence intervals. Linear regressions with fixed effects included a dummy variable indicating treatment, and a vector of pre-treatment individual demographic covariates to improve efficiency. Robust standard errors clustered by group accounted for residual covariance on the group level. Two-tailed $p$-values for the average treatment effect were all smaller than 0.001. Note: $***p < 0.001$, $**p < 0.01$, $*p < 0.05$.

working class ($M_{PM} = 3.55$, $SD = 0.27$; $M_O = 3.29$, $SD = 0.17$; $p = 0.10$, n.s.). However, workers in the participatory meetings condition reported more conflict between managers and workers in Chinese society than workers in the observer condition ($M_{PM} = 3.55$, $SD = 0.27$; $M_O = 3.29$, $SD = 0.17$; $\beta = 0.31$, $CI = [0.21, 0.42]$, $SE = 0.05$, $p < 0.001$).

*Participation outside of work.* Workers in the participatory meetings condition reported higher levels of participation behavior outside of work, averaged across both indices ($M = 4.39$, $SD = 0.19$) than workers in the observer condition ($M = 4.21$, $SD = 0.19$), $\beta = 0.18$, $SE = 0.05$, $CI = [0.03, 0.21]$, $p < 0.001$ (Fig. 1). Workers in the participatory meetings condition reported significantly higher interest in participation in politics ($M = 4.06$, $SD = 0.32$) than workers in the observer conditions ($M = 3.80$, $SD = 0.34$), $\beta = 0.29$, $CI = [0.13, 0.45]$, $SE = 0.08$, $p < 0.001$). Likewise, workers in the participatory meetings condition reported significantly more participation behavior in family and social life ($M = 4.54$, $SD = 0.19$) than workers in the observer condition ($M = 4.41$, $SD = 0.21$), $\beta = 0.12$, $CI = [0.03, 0.21]$, $SE = 0.05$, $p = 0.012$.

**Attitudinal changes toward authority and justice in study 2.** *Generalized attitudes toward authority*: The mean score of generalized attitudes toward authority for the whole sample of 172 U. S. university administrative staff members was 2.77 ($SD = 0.68$) on a 7-point Likert scale. On average, workers tended to "slightly disagree" with statements asserting complete obedience and respect for authority without question. Replicating Study 1,

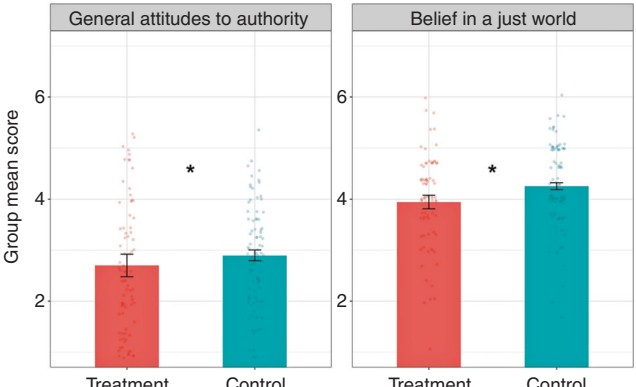

**Fig. 2 Study 2 group mean ratings.** Study 2 group mean ratings of general attitudes toward authority and belief in a just world on a 7-point scale. Ratings are shown as a function of assignment to participatory meetings condition (treatment). Error bars show 95% confidence intervals. Linear regression with fixed effects included a dummy variable indicating treatment, and a vector of pre-treatment individual demographic covariates to improve efficiency. Robust standard errors clustered by group accounted for residual covariance on the group level. Two-tailed $p$-values for the average treatment effect on attitudes toward general authority and justice were 0.037 and 0.045 respectively. There was a joint significance of the average treatment effects on both attitudinal outcomes between administrative staff groups in the participatory meetings condition and the control condition, $F_{1,26} = 3.35$, $p = 0.039$. Note: $***p < 0.001$, $**p < 0.01$, $*p < 0.05$.

workers in groups who experienced participatory meetings reported significantly lower deference toward authority ($M = 2.70$, $SD = 0.88$) than workers in the control condition ($M = 2.90$, $SD = 0.46$; $\beta = -0.44$, $CI = [-0.85, -0.03]$, $SE = 0.21$, $p = 0.037$) (Fig. 2).

*Belief in a just world*: The mean score of belief in a just world for the whole sample was 4.05 ($SD = 0.39$). This value is slightly above the "neither disagree or agree" point of the scale, which indicates that, on average, staff members tended to be neutral with statements asserting generalized justice. As predicted, staff groups in the participatory meetings condition reported significantly lower score in belief in a just world ($M = 3.93$, $SD = 0.41$) than staff groups in the control condition ($M = 4.18$, $SD = 0.35$; $\beta = -0.23$, $CI = [-0.45, -0.005]$, $SE = 0.11$, $p = 0.045$).

As predicted, there was a joint significance of the average treatment effects on both attitudinal outcomes between administrative staff groups in the participatory meetings condition and the control condition, $F_{1,26} = 3.35$, $p = 0.039$. We conclude that participatory meetings significantly changed administrative staff members' generalized attitudes toward authority and justice compared with those in the control condition.

Figure 3 compares the effect sizes for Study 1 and 2. Notably, they are quite similar in size, and understandably Study 2 features larger confidence intervals, due to its smaller sample size. In both Study 1 and 2, participatory meetings changed participants' generalized attitudes toward authority and justice such that treatment workers registered as less authoritarian on a traditional scale of authoritarianism and on widely-used measures of belief in a just world, four (Study 1) or two (Study 2) weeks after the participatory meetings ended.

## Discussion

Across two field experiments, the results provide support for our hypothesis that short-term participatory experiences at a work group could change attitudes that are traditionally conceptualized as stable and a product of one's personality and long-term social

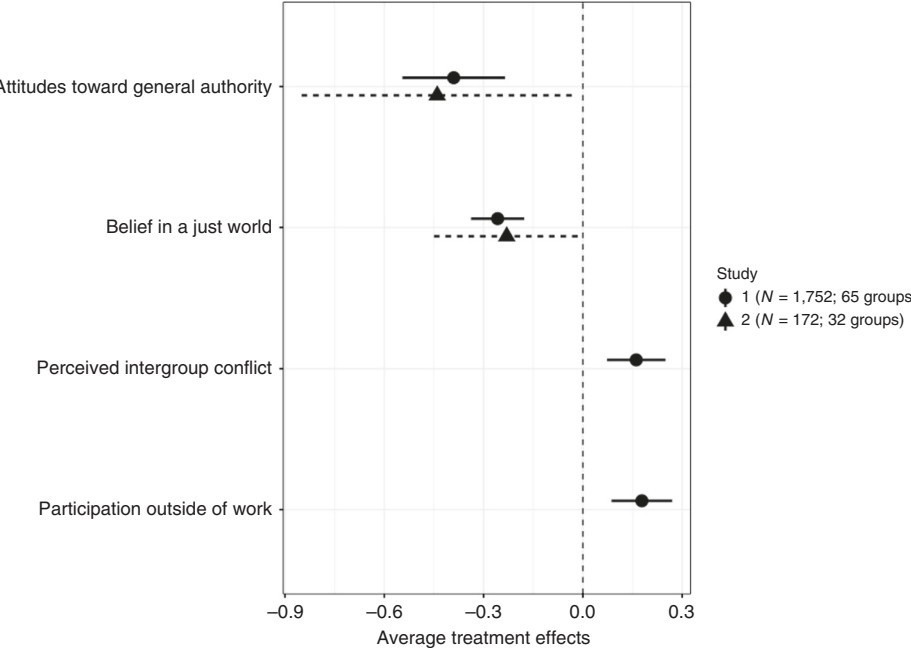

**Fig. 3 Average treatment effects from participatory meetings on social outcomes.** The dots indicate the estimated coefficients for the average treatment effect from linear regressions with fixed effects, with robust standard error clustered on the level of groups. Error bars show 95% confidence intervals. Participatory meetings significantly reduced workers' authoritarian attitudes and belief in a just world, significantly increased perceived intergroup conflict between managers and workers in the Chinese society and workers' participation outside of work.

experience: generalized attitudes toward societal authority and justice. Following a six-week period in which workers experienced a participatory meeting in which workers talked and supervisors listened, treatment workers reported less deference to generalized authority and lower belief in the just nature of the world, compared with workers who were not assigned to these meetings. The size of these changes is roughly one standard deviation on the survey measurement scale, and is perhaps more impressive considering that they were observed one month (2 weeks, in Study 2) after participatory meetings had ended. Generalized attitudes toward authority and justice changed without a shift in larger societal procedural justice and without a shift in the actual mechanisms of the workplace's authority and justice structures, beyond a 20-minute meeting each week. The data suggest that generalized attitudes toward authority and justice can be affected in an enduring way by brief but immersive experiences with a more egalitarian local power structure.

In Study 1, workers assigned to the participatory meetings also reported a heightened level of perceived conflict between managers and workers in the larger Chinese society, but not greater conflict between other dominant and lower status groups, a focused finding that counters a potential alternative interpretation that treatment workers simply reported more negative attitudes. These same workers assigned to participatory meetings also endorsed a more favorable attitude toward their own local factory management, reporting at significantly higher rates that factory management (i.e., the supervisors of their supervisors) cared about and respected them[36]. This effect was observed one week after the intervention and remained robust when re-tested one month later along with the variables reported in this study. Workers in Study 2 did not become more positive about their local authority, likely because of a ceiling effect in positive attitudes toward the university.

The contrast between treatment factory workers' increased positive attitude toward their local authority and their increased critique toward broad societal authority suggests that attitudes toward local and societal authority were differentially influenced

by the intervention. The literature of authority and procedural justice would predict the former[37], but has little to say about the latter. Specifically, authorities who listen to subordinates are perceived as fairer and more legitimate, but the literature on procedural justice does not address whether these interventions affect attitudes toward authority writ large, or authoritarianism. Perhaps, after experiencing a certain amount of voice and participation in the workplace meetings, workers felt more positively about their local workplace authority, a feeling that contrasted with other broad types of authority in their society. Another possibility is that treatment workers' positive experience expressing their voice in front of a local authority changed their expectations for their relationship to authority more generally, and encouraged them to adopt a more critical attitude toward authority and justice in society. Either or both of these related possibilities may be at work in this experiment, and warrant further exploration.

Furthermore in Study 1, treatment workers' self-reported participation in politics and family life also increased, and is of interest to a broader story of change. Workers reported greater engagement with political news and greater participation in family decisions and peer interactions. These findings should be interpreted within the context of the experiment—many of these Chinese workers experience familial and social isolation as migrant workers, and as young women are relatively disempowered even in decisions about their own children that they left back home in rural areas with family[35]. That greater critique of authority and lowered belief in the justness of the world would accompany a self-reported increase in assertion of the self suggests a global shift that might be cautiously labeled "empowerment."

China is home to an authoritarian political system, and Study 1 results could be interpreted as an effect of high contrast—a democratic-style meeting held in an authoritarian society. However, we observe similar findings in Study 2, in which the workplace and societal contexts are drastically different from a hierarchical factory environment in a non-democratic state. The

decreased belief in generalized authority and justice in American university staff groups is particularly intriguing, considering that they are situated in a liberal university in a Western democratic society, and are routinely engaged in active participation in their familial, social, and political life. Indeed, the average level of authoritarianism for all Study 2 administrative staff was low—few individuals reported unconditional deference to authority. Yet the participatory meetings decreased treatment staff's reported authoritarian attitudes further. Thus, it might not be the relative difference in worker participation that drives the attitudinal change toward authority and justice, but rather the implementation of regular opportunities to speak up in group, even in the short term. This regular opportunity to voice out one's opinion is the common factor between these two drastically different settings.

By contrast, what was structurally common for the participants of both experiments was their position as support workers who are lower-positioned in an organizational hierarchy. Both the factory workers and university staff members labor under supervision of direct managers and consider their role to be one of support (to the factory or to faculty and students, respectively). These roles are likely less often appreciated and recognized in their local organizational contexts; inviting and recognizing the voice of workers in these roles might be particularly empowering. Future research should test whether a lower status structural position is necessary for a notable treatment effect to emerge.

Participatory interventions have been popular in developing country settings, where aid organizations have attempted to found new, citizen-driven local institutions and encourage citizen involvement[38]. These interventions have not always been successful[39], but we do not think that they parallel the work we present here. The primary way in which the present intervention is different is that the participatory meetings are a modification to people's every day work environments, as opposed to an invention of a new institution with which no one yet identifies. Our participants already shared an identity with their fellow work group members, met with them every day, and worked within conditions that were relatively less participatory than the treatment period. The contrast between citizen-driven development programs and this participatory meeting highlights our theoretical interpretation that a regular, immersive modification of an individual's everyday world and social groups is the key to changing attitudes toward authority and justice.

We believe that our results provide some of the only causal evidence supporting the theory of participatory democracy[6], which posits that the workplace provides a training ground for the development of democratic attitudes, including attitudes toward authority and ideas about the just or unjust nature of the world. (We imagine this may also be true for schools, which would be a further interesting direction for research.) Underlining the importance of these findings, many contemporary theories of democracy argue that democratic attitudes are necessary for stable democracy[6,40,41]. Attitudes toward authority and justice are likely one set of related attitudes that could cultivate participatory behavior in the civic and political realms. Future work will need to verify participants' self-reported claims about participation by measuring concrete behaviors, perhaps with standardized behavioral games or, with difficulty but high payoff, observed behaviors in the context of these workers' own lives.

Our findings also speak directly to Lewin's original intuition about the creation of democratic spaces within society, and more broadly his idea of a "cultural island," in which group-based cultures develop in compartments that are sometimes separate from that of the larger society[7]. In their work on changing leadership patterns from autocratic to democratic, Lewin and colleagues noted potential conflicting norms and value systems between the face-to-face groups they theorized about and the larger societal setting. In their view, the democratic dynamics of local face-to-face groups could be cultivated apart from a hierarchical society. Our findings support their theories about how certain procedures could cultivate more participatory group members; contra their expectations, we find that the local experience of participation spilled over into workers' attitudes toward the broader world. Our data suggest that these participatory groups were not, in a strict sense, cultural islands.

One might relate theories of participatory democracy to the vibrant literature on deliberative processes[42,43]. We speculate that different mechanisms and outcomes will be involved in participatory or deliberative processes, depending on the goals (e.g., work discussions without decision-making versus decision deliberations prior to voting). Most crucially, deliberative processes tend to take place outside the work contexts, in support of a political decision, while participatory processes within workplaces are theorized to spill over from local workplace to the general society.

Compared with the wealth of previous research on the influence of individual differences on attitudes toward general authority and justice, relatively little research has focused on the local situational factors that influence such attitudes. Our findings support the wisdom of earlier political theory on the spillover effect of local workplace participation. Perhaps surprising is that an entire workplace overhaul may not be the minimum change necessary to influence workers' outlook on society. Our research suggests that a temporary change in experience in individuals' work life can have a modest but weeks-long enduring impact, on social views considered so stable that they are often described as personality traits. Echoing Rousseau and Pateman, future research can explore whether local participatory experiences can not only change general attitudes, but also cultivate a more participatory democratic norm and active citizen engagement in the civil society.

## Methods
**Study 1 group randomization.** We conducted Study 1 at a multinational textile factory in China. We sampled all work groups ($N = 65$) from the factory's sewing departments where workers are organized in groups. Employees in the sewing work groups work on their own tasks, which are related to their coworkers' tasks. For example, one worker may be in charge of sewing the sleeves of a hoodie while another is in charge of sewing the hood pieces. Each work group has its own supervisor who oversees group members' work. The factory requires all work groups to have a 20-minute morning meeting before the start of each workday, in which the supervisor summarizes the previous day's work performance, recommends individual and group work strategies, and announces goals for individual workers. Workers rarely transfer to a different group after they are hired. Individuals in all groups provided informed consent during a recruitment phase one month before the experiment's commencement. We made oral public announcements in the sewing departments to invite all sewing workers to a study called "worker experience in the factory," with the cooperation of the factory's human resource department. Workers were specifically told that "researchers are not part of the factory but are coming to learn management practices and offer new technologies on work-related issues. All of you are invited to take part[…]Participation is completely voluntary."

We randomly assigned the 65 work groups ($N_{workers} = 1752$; 93.6% female; mean age = 32.5 years, ranging from 18 to 53) to participate in a weekly morning participatory meeting (referred to as *participatory meetings condition* or *treatment condition*), or to have an observer attend the usual morning meeting (referred to as *observer condition* or *control condition*) once per week for six weeks. To randomize, we used a non-bipartite matching scheme[44] (see Supplementary Note 3 for matching procedure and code). The building structure allows for little communication between sewing departments and among work groups. Workers spend most of their time in their own group's working area on the production floor during work, and have little communication with other groups during and after work. Thus, we have few concerns about spillover of treatment to control groups.

**Study 1 experimental procedure.** Experimental manipulations were implemented once per week for six weeks in the 20-minute status quo morning meeting slot. Eleven research assistants (RAs), all female graduate students from a local university, were trained by the first author to either facilitate the weekly participatory

meeting or observe status quo meetings led by supervisors, following a detailed experimental protocol (see Supplementary Note 1). RAs were unaware of research hypotheses. During the 6-week experimental period, treatment groups experienced six weekly participatory meetings and control groups experienced six weekly meetings with an outside observer.

*Observer condition (control)* For each control group, a research assistant conspicuously monitored six 20-min status quo meetings during the experimental period. The RA described herself as part of a visiting research team studying management strategies from the production floors. Once per week for 6 weeks, she silently observed and took notes as supervisors led the status quo morning meetings, which were typically 20-min lectures on the group's production performance and working strategies, with zero group participation or discussion. The meetings ended with the supervisor writing goals for each individual worker, in terms of the number of pieces to complete, on a whiteboard where all group members could see. RAs did not encourage any change in the status quo meeting structure.

*Participatory meetings condition (treatment)* For each treatment group, a research assistant facilitated six weekly 20-minute group discussions, followed by an invitation to workers to voice their own goals in front of the group. Just as in the control condition, the RA described herself as part of a visiting research team studying management strategies. The RA then provided prompts to discussion (derived from the most common points supervisors made in previously observed status quo meetings) and encouraged all members of the group to participate in the discussion and the goal setting in the supervisor's presence. Supervisors were informed in advance that they should refrain from speaking, in particular from interrupting the workers. All group members were encouraged to share work experience and production strategies for their own task, such as how to prepare piecework, where to put finished or unwanted pieces, or the best way to pass finished pieces to the next worker in the group. The RA was trained to re-direct non-work-related discussions, and to set a clear expectation of active group participation at the start of the first treatment meeting by saying:

"We encourage everyone to speak up. Say whatever's on your mind about your work, such as issues yesterday or in the past week, the difficulties you have at work, or things you think will help you and others. I may ask some questions, and there are no right or wrong answers. Whatever you share will be helpful for us and for the group."

Following the group discussion, the RA announced the week's order information so that workers could use this information to form their individual production goals for the week. The participatory meetings ended with each group member announcing her goal to the group in terms of specific number of pieces she would like to complete for that week.

Based on the RAs' qualitative field notes and the first author's daily observation, we concluded that workers in the participatory meetings were not given novel work information and did not supply novel work information due to the discussion, relative to workers in the control group with the status quo supervisor meetings. The discussion prompts used by the RAs in participatory meetings were designed based on previous supervisor lectures in status quo meetings and were focused solely on work; moreover, due to the routinized nature of the sewing tasks, there was relatively little room for information variance and novelty in the participatory discussions. Thus, the informational structure of the control status-quo meetings and treatment participatory meetings was largely the same: both were centered around production, work strategies, and had a goal-setting segment at the end of the meeting. The major difference between conditions, we believe, was that the supervisors' voices dominated in the control condition, while the voices of workers dominated in the participatory meetings treatment condition.

**Study 1 data collection**. Four weeks after the experimental intervention had ended, the same team of RAs and the first author collected individual surveys from all 1752 participants in the study's 65 work groups (for procedure see Supplementary Note 8).

The survey (completion rate = 84.07%; 93.49% female) consisted of four parts: generalized attitudes toward authority (e.g., "Obedience and respect for authority are the most important virtues children should learn"), belief in a just world (e.g., "By and large, people deserve what they get"), perceived conflict between different social status groups (e.g., "In your mind, to what extent do the rich and the ordinary have conflict with each other"), participation behavior outside of work (in politics, e.g., "How often do you follow news about politics?"; in family and social life, e.g., "How often have you participated in your family's decision making lately?"). Survey items were adapted from established measurement scales from psychology and were measured with a 6-point Likert scale (see Supplementary Note 8 for the full scale). Both exploratory and confirmatory factor analyses supported these pre-registered item groupings. Questions about demographics (age, gender, marital status, and rural or urban origin) were measured previously, one week after the end of the intervention. All items were translated and back-translated into Mandarin Chinese by two English-Chinese bilingual speakers, and were piloted with an independent sample of factory workers.

In Study 2, we replicated the Study 1 paradigm with administrative staff groups working at a prestigious private university in the United States. Study 2 tests whether groups of knowledge workers in a Western democratic society also react to the relatively small adjustment to their experience of authority in the work place,

similar to workers in more task-based settings in a strictly hierarchical, non-Western and non-democratic environment.

**Study 2 group randomization**. We randomly assigned administrative staff groups to participate in a weekly morning participatory meeting (*participatory meetings* or *treatment condition*) or continue with their status-quo meetings (*control condition*). Thirty-two academic departments' administrative staff groups or 172 individual staff members participated in the study (78% female, 22% male; 80% identified as White or European-American; mean age = 50 years, ranged from 25 to 88 years).

Each group was comprised of an academic manager (the supervisor) and staff members who directly report to the manager (the workers). Supervisors and workers provided written informed consent before the experiment's commencement. The median size of the administrative groups was 6. The staff members' work is relatively independent, including job roles such as graduate and undergraduate administrators, finance managers, and event coordinators.

**Study 2 experimental procedure**. In the participatory meetings condition, supervisors facilitated a weekly 20-minute group discussion on work challenges and goals in each individual department, following a participatory meeting protocol that we designed as a very close translation of the Chinese participatory meeting protocol (see Supplementary Note 1). The main departure was that supervisors had to lead the meetings, for logistical reasons. All supervisors in the treatment condition attended a training on how to use the protocol, and specifically how to encourage their staff to discuss their work challenges, strategies, and goals, while showing active listening and without interrupting. In the control condition, supervisors continued with their status-quo meetings, which we did not attempt to modify. No observer was sent to the control groups to observe their regular meetings. The study design, hypotheses, and analyses are pre-registered at https://osf.io/fj4sr/.

**Study 2 data collection and analysis**. Two weeks after the experiment ended, RAs collected individual surveys from treatment and control workers. Participants from each work group completed a paper survey sitting together in the conference room of their department. We used the same items from Study 1 to measure generalized attitudes toward authority and belief in a just world, except that Study 2 used a 7-point Likert scale (1 = strongly disagree to 7 = strongly agree) while Study 1 used a 6-point scale. We used a 6-point scale and excluded the neutral option for our Chinese sample because from previous studies, we found that Chinese participants were particularly inclined to select "neither disagree or agree," possibly due to the predominant cultural value of the *Middle-Way*[45]. Due to survey time constraints in Study 2, we did not measure perceived intergroup conflict or participation behavior outside of work in Study 2. Like Study 1, we measured the groups' productivity and work-related attitudinal outcomes, which are reported separately. We use the same analysis model as in Study 1. Post-hoc power analysis indicates that the achieved power given our sample size and average effect size was 0.82.

We have complied with all relevant ethical regulations for work with human participants. Informed consent was obtained prior to study implementation. Both studies were approved by Princeton University's Institutional Review Board.

**Reporting summary**. Further information on research design is available in the Nature Research Reporting Summary linked to this article.

## Data availability

Our preregistration, replication code (in R), dataset, and study materials (including surveys in both English and Mandarin Chinese and the training guide for research assistants) can be found at the Open Science Framework (https://osf.io/d9fnh/). R Studio (version 3.6.1) is used for data analysis.

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

## Acknowledgements
The study was approved by Princeton University Institutional Review Board and was funded by grants from the Canadian Institute For Advanced Research to ELP and Princeton University's Fund for Experimental Social Sciences to SJW. We thank Dale Miller, Tim Wilson, Eldar Shafir, Susan Fiske, Peter Aronow, and four reviewers for critical comments and suggestions.

## Author contributions
S.J.W. and E.L.P. developed the study concept and contributed to the study design, data collection and analysis, and manuscript drafting.

## Competing Interests
The authors declare no competing interests.
