## [Peer Review File · Nature Communications]

Reviewers' comments:

Reviewer #1 (Remarks to the Author):

In this paper the authors present two extremely impressive field studies that show how participatory governance meetings at work v. business as usual meetings differentially shape attitudes toward authority and justice such that participation elicits lower endorsement of general obedience, lower belief in justice, greater conflict with management and workers (study 1), and more participation in politics outside of work (study 1).

My impression of the methods and approach here is that it is appropriate and exemplary. The studies are pre-registered, the manipulation is immersive (rather than momentary), and the outcome measures are solid in terms of construct validity.

In terms of the interpretation of the findings, the dichotomy with authoritarianism as a trait-based construct didn't really resonate with me. Some would argue that nearly everything is heritable (see Turkheimer) and that is perhaps immaterial to the contribution of this work--things can be both heritable and situational and we need not believe it goes one way for the other to have meaning. In my view, the contribution of this work is that it seeks to better articulate the everyday situational circumstances that shape attitudes toward authority and justice. That we might not think about everyday organizational contexts as really part of political discourse and in fact, people often try to keep politics out of the office or the college or the research (see Haidt for instance). Here I think what we might argue is that in fact, work contexts can be incredibly meaningful for how you think about big societal constructs like justice and fairness.

I think this (above) framing works a bit more appropriately than the one currently offered because it is also the case that researchers do momentarily shift attitudes toward authority and justice all the time in social psych and poli-sci. Just this week a psych science paper found the exposure to rising inequality increased support for authoritarian leaders (Jetter and colleagues). Warnings about increases in demographic diversity make Whites more conservative and more obedient of authority (Craig & Richeson). The list goes on and on. But what is critical here to me is that workplace structures can set you up for shared governance and then people apply those broad work adopted principles to the world. That is an important insight here that I hope the authors will highlight.

I also thought that there was greater opportunity to speak more about culture (and not just cultural islands in the Lewin case). There are class and country based cultures here that reasonable people might contend would differentially shape the effects here and yet you get consistent findings. It's worth thinking what is structurally shared about these two contexts, and I'm thinking about the low status positions of the workers in China and the US and how that vantage point reacts similarly to shared governance.

Last point is that I do have a little trouble making the leap to why just world beliefs might change based

on this manipulation. Much easier to see how obedience to authority becomes less valued when you have a taste for governance. But, how do big concepts like justice get tripped up. It feels like there is another mediator in that process, as in I can govern, why am I not governing, maybe that is unfair? Or perhaps it is something else but whatever it might be I think explaining this makes sense.

I hope these comments are helpful as you continue to pursue this research!

Michael Kraus

Reviewer #2 (Remarks to the Author):

Wu and Paluck report the results of two field experiments in which the participatory experience of workers was changed by assigning workers to 20-minute participatory meetings, rather than a status-quo meeting. The participatory treatment led to changes in survey responses about authoritarianism, social-justice, and self-reported participation in decision-making.

As the authors nicely argue, this finding makes important contributions to the relationship between social and economic structures and attitudes toward authority. The statistical analysis is appropriate for the type of inferences the authors are attempting. The replication is a particularly nice feature of the experiment because it overcomes some of the concerns about mechanism that are the largest shortcomings of the paper.

I see two primary shortcomings of the paper:

1) It is impossible to separate the content of the messages provided in participatory meetings from the act of participation itself. As I read this manuscript, the authors want to argue that the act of participation itself causes these outcomes. This may be true but consider the counterfactual where the messages delivered in the status quo supervisor-led meeting were the same as the those delivered in the participatory meeting. Would we see the same change in attitudes? The authors imply that the answer is no, but we don't actually know because the messages were not held constant between the conditions. Without a way to separate those, one could argue that the workers could simply have had messages mailed to them in a letter at home, without the participatory element, and the results would have been the same. One could imagine further experiments where the messages are somehow randomly varied, even if the type of participation cannot be randomized at the same time, and the authors could then triangulate these results to shed light on the causal mechanism. However, given that this might not be feasible, I would at least like the authors to acknowledge that this treatment is bundled in that the content of the messages and the mode of delivery cannot be separated.

2) In the first experiment especially, I worry about this inability to separate these mechanisms because, even though the graduate-student leaders were blind to the hypothesis, we might worry that they

would still inject their opinions into the group when given the opportunity and these opinions might be especially anti-authoritarian. This is less of a concern in the second experiment because the supervisor leads both sessions (rather than just observing in a condition as in the first experiment), but again, what if the supervisor changes messages across the conditions?

Again, these shortcomings are not fatal, but the manuscript would be strengthened if this ambiguity around the mechanism of treatment is explored and/or discussed.

Reviewer #3 (Remarks to the Author):

The authors examined in two studies whether creating participatory spaces at the workplace would shift workers' general feelings about authority and justice. The authors examined an interesting, important, and provocative question. The theoretical background of the paper is well articulated, and the interdisciplinary integration of psychological and political theory concepts is laudable. The methods directly test the question of interest, and the analyses are well-executed and described. Overall, I have a very positive evaluation of this paper and believe that it would stand as a generative contribution to multiple academic literatures (in addition to having translational value). I outline several comments below that I would like to see the authors address.

1.) Authoritarianism, when conceptualized as a relatively stable construct, is generally considered to be motivational in nature. Specifically, it is not simply that people who are high on authoritarianism respect authorities, but instead they want and desire to do so. In turn, people high on authoritarianism report negative attitudes and intolerance toward groups that wish to change current structures of authority and hierarchy within a society (e.g., people who embrace a feminist ideology). Linking this to the current studies, the act of participatory democracy is itself subversive to structures of authority and hierarchy, and so it is likely that people who begin the study high on authoritarianism should find the experimental condition to be highly aversive (and could potentially react against it). In other words, the theoretical exposition that the authors currently lay out would seem to lead to the prediction that their manipulation would only be effective for people who begin the study "lower" on support for authority. I would encourage the authors to address this point in two potential ways:

The first is to clarify whether they think support for authority is a motivational construct. Based on my current reading of their introduction, I would think this is the case. However, if they do not believe that they are assessing a motivational construct, then they should lay out their conceptualization accordingly and situate/reconcile their explanation within the broader literature that is currently motivationally focused.

The second approach is to conduct secondary/exploratory analyses to determine whether the manipulation might have differentially affected people who are at baseline "higher" or "lower" on support for authority. I realize this is not a pre-post-test design and so the analysis could not directly test this, but the authors do have proxy variables that could work. For example, men and older people tend to be higher in support for authority than women and younger people, and the authors could examine if these demographic variables moderate the strength of their manipulation (when there is enough

variability in the construct to test for moderation, of course).

2.) The authors draw in part from political theory concerning participatory practices to generate their predictions. However, there is a related literature examining deliberative practices (e.g., Habermas' conceptual ideas and researchers' subsequent empirical work). This concept in some ways overlaps with participation, yet is distinct in the proposed mechanisms. Participation as a process has (to some degree) tended to focus on the generation of social bonds, whereas deliberation has focused more on individual knowledge building. Both processes could lead to the observed effects. Given that the authors do not examine mechanisms of the experimental manipulation on the outcome variables, it would be helpful if the authors could speculate about the potential role of mechanisms derived from both participatory and deliberative perspectives.

3.) In both studies the standard deviations in attitudes toward authority and justice tend to be strikingly low. It would be helpful if the authors could comment on why this might be and if it has any implications for the interpretation of their findings.

Reviewer #4 (Remarks to the Author):

I want to commend the authors of this manuscript on two features of their work. First, it is extremely difficult to implement a randomized treatment to workgroups in organizations, so that aspect of their research is one significant contribution. This is particularly true for understanding the effect of worker participation on political attitudes because the work in this area suffers from endogeneity issues. Second, they have convergent evidence from two experiments in two different countries. All of this reflects a very ambitious research agenda. The designs of the studies, and their execution, appear to be very carefully crafted, judging by the main manuscript and the supplementary materials. The analyses are also appropriately sophisticated. In other words, I am impressed with the technical execution of this research.

From a theoretical perspective, I understand that the authors are more trying to test classic psychological theory, to see if it holds up in the field, than they are trying to contribute to that theory. To the extent that the findings are surprising, it is not about the direction of the effects but that there is any effect at all of democratizing one's workgroup on political attitudes. But the theorizing is very impoverished here, and I was surprised by this given the sophistication of the empirical part of the project. I am a big fan of Lewin but the theorizing is based almost entirely on Lewin (in addition to Pateman), but theorizing is generous here because it's not clear what the mechanism is supposed to be. I'd like to hear more from you about what you think is going on, what are the necessary and sufficient conditions in your treatment, why do the dynamics of one's workgroup at work affect one's broader politically-related attitudes? On the dependent measure side, it wasn't entirely clear why you measured exactly what you measured. For example, the belief in a just world did not seem like an obvious choice, and I don't see your justification for including it. Similarly, why look at perceptions of intergroup conflict?

I want to also mention that I appreciate the synthesis of social psychology and political science here, but

there is still a ton of literature missing. The authors seem to be open to a broad interdisciplinary base, and one cannot avoid some blind spots, but ultimately this is a paper about organizational behavior and political attitudes, yet research on management and organizations is mostly missing. I'm thinking that by digging into the literature on organizations, particularly the literatures on trust and/or justice, the authors might develop their thinking further about what mechanism(s) are operating in their settings. I would broaden your review on the dependent variable side as well. What other effects (e.g., other work attitudes) have been found from workgroup democratization, worker participation, quality circles (an organizational structure that has some of the democratic properties that interest you), etc.? There is also more literature in political science and sociology about democracy at work, and I would recommend looking at the literature on unions and union commitment or participation. All of these literatures have the potential to enrich your theorizing in the front end of the paper. I had to do some digging to find many of the citations below, but I did so in the spirit of improving this manuscript because it has a lot of promise.

To be clear, the causal inferences that we can make based on the authors' current research are much clearer than any of the work I recommend they look at below. That is, the research that they have done is very good, but the manuscript needs a lot of work. I wonder too about whether it is a good fit for this publication outlet, but I leave that to the editors. To the authors, if you frame the paper more in organizational terms, I could see it having an audience at an outlet like Management Science.

Workplace Participation References

Adman, Per. "Does workplace experience enhance political participation? A critical test of a venerable hypothesis." *Political Behavior* 30.1 (2008): 115-138.

Carter, Neil. "Political participation and the workplace: The spillover thesis revisited." *The British Journal of Politics and International Relations* 8.3 (2006): 410-426.

Greenberg, Edward S., Leon Grunberg, and Kelley Daniel. "Industrial Work and Political Participation: Beyond" Simple Spillover"." *Political research quarterly* 49.2 (1996): 305-330.

Edward Greenberg's book "Workplace Democracy" about cooperatives in the Pacific Northwest United States.

Timming, Andrew, and Juliette Summers. "Is workplace democracy associated with wider pro-democracy affect? A structural equation model." *Economic and Industrial Democracy* (2018).

Elaine Bernard's chapter "Creating Democratic Communities in the Workplace" in Mantsios' edited volume "A New Labor Movement for the New Century" (1998)

Weber, Wolfgang G., Christine Unterrainer, and Thomas Höge. "Psychological Research on Organisational Democracy: A Meta-Analysis of Individual, Organisational, and Societal Outcomes."

Applied Psychology (2019).

Estlund, Cynthia. *Working together: How workplace bonds strengthen a diverse democracy*. Oxford University Press, 2003.

Sobel, Richard. "From occupational involvement to political participation: An exploratory analysis." *Political Behavior* 15.4 (1993): 339-353.

Steel, Robert P., and Russell F. Lloyd. "Cognitive, affective, and behavioral outcomes of participation in quality circles: Conceptual and empirical findings." *The Journal of Applied Behavioral Science* 24.1 (1988): 1-17.

Some examples of related research on unions.

Kerrissey, Jasmine, and Evan Schofer. "Union membership and political participation in the United States." *Social forces* 91.3 (2013): 895-928.

Terriquez, Veronica. "Schools for democracy: Labor union participation and Latino immigrant parents' school-based civic engagement." *American Sociological Review* 76.4 (2011): 581-601.

Reviewer # 1 comments:

1. In terms of the interpretation of the findings, the dichotomy with authoritarianism as a trait-based construct didn't really resonate with me. Some would argue that nearly everything is heritable (see Turkheimer) and that is perhaps immaterial to the contribution of this work--things can be both heritable and situational and we need not believe it goes one way for the other to have meaning. In my view, the contribution of this work is that it seeks to better articulate the everyday situational circumstances that shape attitudes toward authority and justice. That we might not think about everyday organizational contexts as really part of political discourse and in fact, people often try to keep politics out of the office or the college or the research (see Haidt for instance). Here I think what we might argue is that in fact, work contexts can be incredibly meaningful for how you think about big societal constructs like justice and fairness.

I think this (above) framing works a bit more appropriately than the one currently offered because it is also the case that researchers do momentarily shift attitudes toward authority and justice all the time in social psych and poli-sci. Just this week a psych science paper found the exposure to rising inequality increased support for authoritarian leaders (Jetter and colleagues). Warnings about increases in demographic diversity make Whites more conservative and more obedient of authority (Craig & Richeson). The list goes on and on. But what is critical here to me is that workplace structures can set you up for shared governance and then people apply those broad work adopted principles to the world. That is an important insight here that I hope the authors will highlight.

- We completely agree with Reviewer 1's takeaway points about the underappreciated central role of workplace contexts in the development of political discourse and attitudes. We feature Pateman and Lewin's theories as motivational background for our hypotheses on page 6; also on page 6 we changed "the attitudinal and behavioral influence of a participatory workplace" to "the political influence of a participatory workplace" to sharpen the prediction and the contribution, as R1 indicates. We also return to this point in our discussion, e.g. on page 22 "We believe that our results provide some of the only causal evidence supporting the theory of participatory democracy (Pateman, 1970), which posits that the workplace provides a training ground for the development of democratic attitudes, including attitudes toward authority and ideas about the just or unjust nature of the world."

We do also think there is something particular about the dependent variables—authoritarianism and belief in a just world—we are measuring here. Past literatures do tend to portray these variables as individual differences, as known through the classic work of Adorno, Rubin, & Peplau, and through the present day work of Pettigrew and colleagues. As Reviewer 1 mentions, there is a great deal of excellent research on whether certain interventions (and situations/contexts) change attitudes toward a particular authority figure or institution. We know of no other study that has found an enduring change in generalized attitudes, a la Adorno and Pettigrew. Thus we believe we are

contributing to an evidential base that is complimentary to but also distinct from the prior literature on induced changes to political attitudes. We sharpen this point in the introduction, making sure to define authoritarianism as generalized attitudes toward authority (not to particular types of authority, or to particular leaders) on pages 4 and 5, and by adding references mentioned by Reviewer 1 on page 5 that show how these changes are complimentary but distinct.

2. I also thought that there was greater opportunity to speak more about culture (and not just cultural islands in the Lewin case). There are class and country based cultures here that reasonable people might contend would differentially shape the effects here and yet you get consistent findings. It's worth thinking what is structurally shared about these two contexts, and I'm thinking about the low status positions of the workers in China and the US and how that vantage point reacts similarity to shared governance.
 - We now discuss further about the structural similarity of the two cultures on page 21: “By contrast, what was structurally common for the participants of both experiments was their position as support workers lower in the organizational hierarchy. Both the factory workers and university staff members labor under supervision of direct managers and consider their role to be one of support (to the factory or to faculty and students, respectively). These roles are likely less often appreciated and recognized in their local organizational contexts; inviting and recognizing the voice of workers in these roles might be particularly empowering. Future research should test whether a lower status structural position is necessary for the treatment effect to emerge...”
3. Last point is that I do have a little trouble making the leap to why just world beliefs might change based on this manipulation. Much easier to see how obedience to authority becomes less valued when you have a taste for governance. But, how do big concepts like justice get tripped up. It feels like there is another mediator in that process, as in I can govern, why am I not governing, maybe that is unfair? Or perhaps it is something else but whatever it might be I think explaining this makes sense.
 - We especially appreciate this point and we now make our rationale behind all hypotheses explicit and concentrated in one space, on page 7: “In this research, we predict that individuals will become less deferent to authority following an immersive group experience in which individuals are encouraged to speak up in front of authority and to assume more authority over their work life. The theoretical intuition is that the feeling of decreased deference will spill over into their assessment of other types of authority. Individuals may increase their appreciation for the specific authorities who facilitate this participatory experience, following the predictions of procedural justice (e.g., Tyler & Lind, 1992). Likewise, individuals may be less likely to believe in a just world, given that the fair and egalitarian procedures of their workplace experience

may provide a contrasting reference point to individuals' evaluation of fairness and equality in the world more generally.”

Reviewer #2 comments:

1. It is impossible to separate the content of the messages provided in participatory meetings from the act of participation itself. As I read this manuscript, the authors want to argue that the act of participation itself causes these outcomes. This may be true but consider the counterfactual where the messages delivered in the status quo supervisor-led meeting where the same as the those delivered in the participatory meeting. Would we see the same change in attitudes? The authors imply that the answer is no, but we don't actually know because the messages were not held constant between the conditions. Without a way to separate those, one could argue that the workers could simply have had messages mailed to them in a letter at home, without the participatory element, and the results would have been the same. One could imagine further experiments where the messages are somehow randomly varied, even if the type of participation cannot be randomized at the same time, and the authors could then triangulate these results to shed light on the causal mechanism. However, given that this might not be feasible, I would at least like the authors to acknowledge that this treatment is bundled in that the content of the messages and the mode of delivery cannot be separated.

- We acknowledge the general point—an intervention that goes beyond a printed one-time message will naturally be bundled with other social stimuli—and we are happy to make this clear in the text. However, we did seek to make the informational content of the treatment and control conditions highly comparable. We pull information up from our supplementary materials to make this clear now, in the main text. Please see our expanded description of the informational structure of the meetings and the role of the research assistants as discussion facilitators, on pages 10 and 11.

Page 10: “The RA then provided prompts to discussion (derived from the most common points supervisors made in previously-observed status quo meetings) and encouraged all members of the group to participate in the discussion and the goal setting in the supervisor’s presence.”

Page 11: “Based on the RAs’ qualitative field notes and the first author’s daily observation, we concluded that workers in the participatory meetings were not given and did not independently supply novel work information, relative to that in the control group with the status quo supervisor meetings. Because the participatory discussion prompts were derived from status quo supervisor lectures, were focused solely on work, and due to the routinized nature of the sewing tasks, there was relatively little room for information variance and novelty. Thus, the informational structure of the control status-quo meetings and treatment participatory meetings was largely the same: both were centered around production, work strategies, and had a goal-setting segment at the end of the meeting. The major difference between conditions, we believe, was that

the supervisors' voices dominated in the control condition, while the voices of workers dominated in the participatory meetings treatment condition.”

2. In the first experiment especially, I worry about this inability to separate these mechanisms because, even though the graduate-student leaders were blind to the hypothesis, we might worry that they would still inject their opinions into the group when given the opportunity and these opinions might be especially anti-authoritarian. This is less of a concern in the second experiment because the supervisor leads both sessions (rather than just observing in a condition as in the first experiment), but again, what if the supervisor changes messages across the conditions?
 - We now clarify the RAs' role and the discussion content (page 10-11). As mentioned in the prior point, the RAs' role was to facilitate the discussion by using question prompts prepared by the authors (see the supplementary materials). RAs only facilitated the work discussion without giving new information and were trained to refrain from talking about non-work related issues. They were also trained to direct non-work related discussion (e.g. food in the cafeteria) back to work if it occurred (page 11). As Reviewer 2 recognizes, in Study 2 supervisors themselves facilitated the participatory meetings in the treatment condition. They were also trained to facilitate the meetings by giving question prompts (see the supplementary materials) and not to inject their own personal opinions. But more to the point, the supervisors of the university administrators would be very unlikely to use the space and time as a platform for discussing anti-authoritarian attitudes. We believe the consistent results from this experiment diminish the chances that this mechanism is responsible for the first but not the second experiment.

Reviewer #3 comments:

1. In other words, the theoretical exposition that the authors currently lay out would seem to lead to the prediction that their manipulation would only be effective for people who begin the study “lower” on support for authority. I would encourage the authors to address this point in two potential ways: The first is to clarify whether they think support for authority is a motivational construct. Based on my current reading of their introduction, I would think this is the case. However, if they do not believe that they are assessing a motivational construct, then they should lay out their conceptualization accordingly and situate/reconcile their explanation within the broader literature that is currently motivationally focused.

The second approach is to conduct secondary/exploratory analyses to determine whether the manipulation might have differentially affected people who are at baseline “higher” or “lower” on support for authority. I realize this is not a pre-post-test design and so the analysis could not directly test this, but the authors do have proxy variables that could work. For example, men and older people tend to be higher in support for authority than women and younger people, and the authors could examine if these demographic

variables moderate the strength of their manipulation (when there is enough variability in the construct to test for moderation, of course).

- We agree with R3 that the concept of authoritarianism is not only a belief structure, but also motivational cognition (we add this point on page 5). We conducted the reviewer's suggested exploratory analysis and found that demographic variables such as gender and age did not moderate the treatment effect. We now clarify this in a footnote on page 14: "Because authoritarian attitudes are motivational in nature, meaning individuals who are high in authoritarianism may not wish to join participatory groups, one might expect a heterogeneous effect that participants who had higher baseline authoritarianism might be less responsive to the intervention. But we do not find a heterogeneous effect using proxy variables such as gender and age for authoritarianism."

We recognize and agree with R3 that this was not a perfect test of the hypothesis, but nonetheless it is the best test with the existing data and so it merits inclusion in the main text. We'd like to recognize Reviewer 3's excellent point, but we feel a reframing of the introduction as assessing the motivational nature of authoritarianism might be misleading as we did not have the data to more directly test this point.

2. The authors draw in part from political theory concerning participatory practices to generate their predictions. However, there is a related literature examining deliberative practices (e.g., Habermas' conceptual ideas and researchers' subsequent empirical work). This concept in some ways overlaps with participation, yet is distinct in the proposed mechanisms. Participation as a process has (to some degree) tended to focus on the generation of social bonds, whereas deliberation has focused more on individual knowledge building. Both processes could lead to the observed effects. Given that the authors do not examine mechanisms of the experimental manipulation on the outcome variables, it would be helpful if the authors could speculate about the potential role of mechanisms derived from both participatory and deliberative perspectives.
 - Again, this is a nice suggestion. We now discuss the potential mechanisms involved in participatory and deliberative processes, on page 23: "One might relate theories of participatory democracy to the vibrant literature on deliberative processes (Habermas, 1989; Fishkin, 2018). We speculate that different mechanisms and outcomes will be involved in participatory or deliberative processes, depending on the goals (e.g., work discussions without decision-making versus decision deliberations prior to voting). Most crucially, deliberative processes tend to take place outside the work contexts, in support of a political decision, while participatory processes within workplaces are theorized to spill over from local workplace to the general society."

3. In both studies the standard deviations in attitudes toward authority and justice tend to be strikingly low. It would be helpful if the authors could comment on why this might be and if it has any implications for the interpretation of their findings.
 - We do not think the size of the standard deviation is strikingly low relative to other datasets with comparable populations—perhaps R3 had some particular standard in mind? Not only do we view the variance as roughly comparable with other populations, we point out that we have a considerable sample size, particularly in Study 1 (65 work groups, $N_{workers} = 1,752$).

Reviewer #4 comments:

From a theoretical perspective, I understand that the authors are more trying to test classic psychological theory, to see if it holds up in the field, than they are trying to contribute to that theory. To the extent that the findings are surprising, it is not about the direction of the effects but that there is any effect at all of democratizing one's workgroup on political attitudes. But the theorizing is very impoverished here, and I was surprised by this given the sophistication of the empirical part of the project. I am a big fan of Lewin but the theorizing is based almost entirely on Lewin (in addition to Pateman), but theorizing is generous here because it's not clear what the mechanism is supposed to be. I'd like to hear more from you about what you think is going on, what are the necessary and sufficient conditions in your treatment, why do the dynamics of one's workgroup at work affect one's broader politically-related attitudes? On the dependent measure side, it wasn't entirely clear why you measured exactly what you measured. For example, the belief in a just world did not seem like an obvious choice, and I don't see your justification for including it. Similarly, why look at perceptions of intergroup conflict?

I want to also mention that I appreciate the synthesis of social psychology and political science here, but there is still a ton of literature missing. The authors seem to be open to a broad interdisciplinary base, and one cannot avoid some blind spots, but ultimately this is a paper about organizational behavior and political attitudes, yet research on management and organizations is mostly missing. I'm thinking that by digging into the literature on organizations, particularly the literatures on trust and/or justice, the authors might develop their thinking further about what mechanism(s) are operating in their settings. I would broaden your review on the dependent variable side as well. What other effects (e.g., other work attitudes) have been found from workgroup democratization, worker participation, quality circles (an organizational structure that has some of the democratic properties that interest you), etc.? There is also more literature in political science and sociology about democracy at work, and I would recommend looking at the literature on unions and union commitment or participation. All of these literatures have the potential to enrich your theorizing in the front end of the paper. I had to do some digging to find many of the citations below, but I did so in the spirit of improving this manuscript because it has a lot of promise.

- Thanks to Reviewer 4 for the excellent suggestions. We now explicitly state the intuition behind the hypotheses and recognize the situations that might explain potential mechanisms of change (e.g. p7, p19, p20, p22). See also above, our similar response to R1's question that shares this theme, point #3.

- Thank you for providing the relevant citations on this broad topic. In the literature review, we now cite more of the important contributions from many social sciences disciplines such as psychology, political science, and organizational science. We also add more discussion of this past work as space allows. We recognize that much of the work is not experimental and offers mixed support for the theoretical hypothesis of participatory democracy (p6-7). We also clarify that Lewin never theorized the spillover of workplace participation to democratic attitudes (p6).
- In cutting the paper down to the word limit of the journal, we perhaps misbalanced the size of the literature review. Our theoretical contribution is to demonstrate that one does not need to completely democratize a culture to change these generalized attitudes. And we think a potential mechanism of change is that workplace contexts serve as a training ground for democratic attitudes to emerge, and provide a reference or contrasting point to which individuals' perceptions of participation in the broader society is compared (this is all more explicitly stated in our theoretical intuition overview, on page 7).

****REVIEWERS' COMMENTS:**

Reviewer #1 (Remarks to the Author):

The major claim of the paper is that an intervention in shared governance for work groups (v. status quo meetings) shapes political attitudes in the realm of both ideas about justice and responses to authority. The work is methodological rigorous and theoretically interesting and my positive opinion of the overall novelty and general contribution of the work is very high.

I read the prior version of this manuscript and I think the authors did an excellent job of handling my comments and those of the three other reviewers. I have no additional concerns at this time. I think the framing is much better situated in the prior literature, especially in terms of participatory democracy.

Reviewer #2 (Remarks to the Author):

The authors have adequately addressed my concerns raised in my initial review. I believe this is a valuable contribution and should be published.

Reviewer #3 (Remarks to the Author):

I feel that the authors have sufficiently addressed all of my (and the other reviewers') comments, and that this paper makes an important contribution to the literature. I would recommend this paper for publication.

Reviewer #4 (Remarks to the Author):

Thank you for revising your manuscript. You tried to respond to most of my comments, and you responded to other reviewers' excellent comments. As a result, your manuscript has improved. Yet, there are two outstanding issues.

1. I don't think you have articulated your reasoning for why you expected a shift in belief in a just world as clearly as you could have.
2. I still don't see any clear rationale for why you would expect a change in perceptions of intergroup conflict. These findings are nuanced and receive some attention in the discussion. But like #1, the manuscript would be stronger if your reason for including the measures in the study in the first place were presented clearly.

****REVIEWERS' COMMENTS:**

Reviewer #1 (Remarks to the Author):

The major claim of the paper is that an intervention in shared governance for work groups (v. status quo meetings) shapes political attitudes in the realm of both ideas about justice and responses to authority. The work is methodological rigorous and theoretically interesting and my positive opinion of the overall novelty and general contribution of the work is very high.

I read the prior version of this manuscript and I think the authors did an excellent job of handling my comments and those of the three other reviewers. I have no additional concerns at this time. I think the framing is much better situated in the prior literature, especially in terms of participatory democracy.

Reviewer #2 (Remarks to the Author):

The authors have adequately addressed my concerns raised in my initial review. I believe this is a valuable contribution and should be published.

Reviewer #3 (Remarks to the Author):

I feel that the authors have sufficiently addressed all of my (and the other reviewers') comments, and that this paper makes an important contribution to the literature. I would recommend this paper for publication.

Reviewer #4 (Remarks to the Author):

Thank you for revising your manuscript. You tried to respond to most of my comments, and you responded to other reviewers' excellent comments. As a result, your manuscript has improved. Yet, there are two outstanding issues.

1. I don't think you have articulated your reasoning for why you expected a shift in belief in a just world as clearly as you could have.
2. I still don't see any clear rationale for why you would expect a change in perceptions of intergroup conflict. These findings are nuanced and receive some attention in the discussion. But like #1, the manuscript would be stronger if your reason for including the measures in the study in the first place were presented clearly.

- We now further clarify the reasoning around our predictions on belief in a just world and intergroup perceptions on page 7: "Likewise, individuals may be less likely to believe in a just world and in hierarchical arrangements of social groups (e.g. of workers and managers), if the fair and egalitarian procedures of their workplace

experience provide a contrasting reference point to individuals' perception of fairness and equality in the world more generally."